# Box–Behnken Design for Polycarbonate-Pigment Blending: Applications and Characterization Techniques

**DOI:** 10.3390/polym14224860

**Published:** 2022-11-11

**Authors:** Jamal Alsadi, Mohammed Taleb Obaidat, Rabah Ismail, Issam Trrad, Marwa Aljamal, Mohammed Dahim, Musab Abuaddous, Mohanad Khodier, Randa Hatamleh, Hashem Al-Mattarneh

**Affiliations:** 1Engineering College, Jadara University, Irbid 21110, Jordan; 2Engineering College, King Khalid University, Abha 62529, Saudi Arabia; 3Engineering College, Yarmouk University, Irbid 21110, Jordan

**Keywords:** experimental design (Box–Behnken design), statistical optimizations, processing at center point (G.T.), micro-Ct scanner, SEM, DOM, FT-IR, spectrophotometer, rheology, dispersion

## Abstract

Incorporating pigments into polymers can be done for various purposes, including the introduction of color, interfacial effects, or aesthetics. If these pigments are to disperse properly, then the process of extrusion must be optimized. During polymer compounding extrusion, three effective processing factors were investigated: feed rate (FR), speed (Sp.) and temperature (temp.) for a colored compounded polycarbonate (PC) grade (30/70%). The processing design techniques were obtained by applying design experiments in a response surface methodology (RSM) to blend two polycarbonates with pigments and optimize the processing temperatures at center points. The first study decided to utilize the response surface approach of Box–Behnken design (BBD) to design an experiment to optimize the process parameters. Statistical significance was demonstrated by the model passing all diagnostic tests. Furthermore, the three processing factors strongly impacted the characteristics of the tri-stimulus color, according to the results from a variance analysis. The second study identified process variables for the same PC grade at the center level, 25 kg/h FR, 750 rpm speed, and (255 °C) temp. The characterization and scanning morphology were examined using MicroCtscanner image analysis, SEM, DOM, rheology, FT-IR, and color-pigmented values were measured using a color spectrometer. The output response was significantly impacted when excellent color dispersion was observed with few agglomerates and less differences in colors at the center point. By characterizing these results and having good insight into color difference output and processing condition relationships, which have an adverse effect on color variation characteristics and minimize recycling compounds of different grades, results in cleaner environments benefits.

## 1. Introduction

Because plastics are so much easier to fabricate than other materials, the plastics sector has grown rapidly in the last several decades, creating plastics in hues that will sell well. Adding color, either for utilitarian or aesthetic purposes, is the primary goal of incorporating pigments into the polymer. Plastic requires color, which may be achieved by mixing in one or more pigments, but achieving the right shade first time can be difficult. In addition to pigment composition, the extrusion process parameters for the polymer compound directly impacts the polymer’s color quality. For example, these factors include the FR, temp., screw configuration, residual time, and much more. In polymer compounds, there have been few investigations into the influence of processing factors on color [1,2]. Pigments can undergo chemical reactions as a result of the processing conditions. If a reaction happens for any reason, the pigments used in the application are thought to have failed. Establishing an accurate color standard using spectrophotometric data is essential for color matching. When it comes to quality control, spectrophotometers are a beneficial instrument for quantifying and defining changes in color. Standard values, such as those established by the CIE, are utilized worldwide when attempting to quantify color [3]. The Commission Internationale de l’Eclairage organization was created in 1913. A color measuring method known as CIELAB is utilized by CIE to measure the values of L*, a*, and b*, respectively. The letter L* represents brightness differences between dark and light, the letter a* symbolizes green/red values, whereas the letter b* represents blue/yellow values [4]. Therefore, the deviation of L*, a* and b* values is represented dE*.
(dE*) = [(Db*)^2^ + (Da*)^2^ + ((DL*)^2^]^1/2^(1)

The color difference magnitude is presented, but the direction of the difference is not. Quality control is carried out by comparing the colored materials produced to a standard. Color disparities concerning goal values in terms of dL*, da*, and db* are utilized rather than absolute values of color. In CIELAB space of color the total change, dE*, is used to represent color differences [5]. 

It has been discovered that integrating additives into polymeric components may have unforeseen effects on their viscosity, mechanical characteristics, and aesthetic appearance [6,7,8]. Adding color to the pure resin has been examined by many scientists. Polymers and mixes, as well as other semi-solid, solid, and fluid materials have all been studied using rheology research. In addition, rheological qualities play a significant role in connecting the various processing stages to the ultimate product state [9,10]. 

Research into the rheology of materials, adjustment to the melt flow rate (MFA) and other literature reviews were the major subjects of this inquiry, including viscosity and hake blender readings [11,12,13].

The present findings indicate compounds may be identified and localized due to good processing. In addition, the rheological properties may be improved by studying chemical groups or chemical compounds that absorb infrared light, which relies on the components or combination of various materials (e.g., fillers, pigments, additives, resins, etc.) [14,15].

Processing characteristics may be improved by adjusting the temperature and the number of additives employed [16,17]. The viscosity of the plastic has been found to influence the color dispersion process in the plastic, which is the crucial outcome of the processing.

As previously observed in sections treated in an previous rheological experiments, adding colorant reduces the viscosity and decreases the absorption mechanisms prior to deterioration and yellowing [18].

The experiment statistical design: the methodology of response surface (RSM) can used to investigate how color change responds to changes in these processing factors. The initial step in this strategy is to appropriately construct tests to assess the model’s parameters when running trials efficiently. After this, a second-order polynomial for color responses can be constructed [19]. Model coefficients can then be predicted by regression of the collected experimental data. The specifics of the defined parameter estimations for the model are disscussed elsewhere [20]. The BBD is defined as an efficient supply of adequate information that evaluates the model’s usefulness. Additionally, it decreases the time and costs required to conduct trials because BBD does not need a significant quantity of design points. This research focused on improving the extrusion process parameters required to achieve polymer pigment dispersion. The color difference was recorded at different color chip samples for the slice exposed to various temperatures (G.T.) and Box–Behnken designs (BBD), all characterized by using a color spectrophotometer and a micro-CT scanner. The pigment dispersion that was created, utilizing the appropriate processing, resulted in a substantial color variation in the sample. This difference was documented based on the grinding temperature. These findings highlight the significance of certain process variables, such as the importance of employing methodical testing strategies when developing optimal processes for various pigment optimization samples and the center point temperatures of the pigment dispersion weathering properties.

Several studies have investigated the breakdown behavior of PC and have been conducted utilizing various diagnostic approaches, such as FT-IR–TGA [21,22,23]. These works explained PC rheology and simultaneous TGA evaluation of the generated volatiles in this procedure [24]. In addition, the processing circumstances of two distinct melt flow indexes (MFI) of polycarbonates were linked to the study’s characterization findings.

All spectra were normalized to be able to compare peak intensities. However, various carbonate specie spectral ranges produce IR active features in the range of 1000–1800 cm^−1^. According to rheological testing, colorant addition to compound plastic reduces viscosity while increasing absorption. Following these results, it is evident that adding colorant lowers viscosity. In addition, degradation and yellowing absorbance processes are reduced, as previously shown in materials previously reported [25]. Several reactions are involved in the PC blend thermal breakdown. Accordingly, its weight loss statistics from the TGA were subsequently reported. These may be used to forecast the kinetics of reactions and offer a notion as to the thermal behavior of polymeric materials throughout pyrolysis, which may be valuable for other reactor model procedures [26,27].

In the presented detail analysis, FR, Sp., and temp. are the most crucial extrusion processing parameters for polymer compounds. Utilizing Box–Behnken design and identifying process variables for the same PC grade at the center level are the goals of this research. The characterization was examined using micro-Ct scanner image analysis, and color values were measured using a spectrometer. Optimization and characterization of processing parameters was conducted to ensure minimum color property deviation while also reducing agglomerations required for processing parameters and minimizing waste to positively impact on the environment.

## 2. Materials and Methods

In this project, we employed a combination of four distinct pigments and two different polycarbonate resins. Table 1 highlights the grade color formulations expressed in parts per hundred (PPH). Resin 1 has a 25 g/10 min melt flow index (MFI), whereas resin 2 had a 6.5 g/10 min MFI. Three additives were used: weather-resistant, light stabilizer and stabilizer. It also included white, yellow, red, and black pigments of color. The composition of one compound grade (R1 30 percent, R2 70 percent) is shown in Table 1.

The processing was performed using a motor twin screw extruder, 27 kW, with a diameter equal to 25.5 mm, a Do/Di = 1.55 and a L/D = 37 ratios. Ten distinct heating zones were included in the extruder, with one on the die and nine on the barrel. As shown in Figure 1, the experimental design considered the threetemp process parameters (FR, Sp., and the heating zone temp.). 

The diagram in Figure 1 explains the compound processing steps. By intermeshing using a Coperion twin co-rotating screw extruder of 25.5 mm (S.B.), the materials were extruded. 0.86% of the total weight was accounted for the pigment and the color additive. The experiment used two PC resins in the proportions of R2 = 70% and R1 = 30%. In a 0.86:100 ratio, the resins were blended with the additives and pigments by a super floater batch; then they molded into three (3 × 2 × 0.1”) color chips, shown in Figure 1, using an injection machine.

The equipment used: compounding extrusion, in order to produce an accurate color, demands that the working conditions be appropriate. In each specimen, at three locations, a spectrophotometer (Figure 2) took measurements of color to obtain L*, a*, and b* tristimulus values. After this, dL*, da*, and db* values were assigned to each color difference. A micro-Ct scanner, SEM and DOM, shown in in Figure 3. (a, b, and c, respectively), characterized the color chips. 

## 3. Results 

### 3.1. Set Up and Experimental Design

#### 3.1.1. Design Variant Processing Parameters (General Trends)

Color was studied in this experiment by varying the operating parameters in a controlled manner.

Three parameters, temperature, speed, and feed rate were varied individually to three levels. While keeping all other parameters fixed (G.T.). Strong interactions were observed between the operating conditions and color.

The experiments were set up as follows: the recommended processing temperatures were = 230 °C, 255 °C and 280 °C with a speed of 750 rpm and a fixed flow rate of 25 kg/h. A similar procedure was used for both the speed and flow rate. The following were recommended: flow rates of 20 kg/h, 25 kg/h, and 30 kg/h, with a constant speed (750 rpm) and temperature (255 °C). Lastly, the focus extended to the recommended processing speeds of 700 rpm, 750 rpm and 800 rpm with a fixed flow rate (25 kg/h) and temperature (255 °C), as shown in Table 2.

Figure 4 shows the effect of temperature on color difference (dE*) for the R1 30% sample that was processed at the center point. It shows a steady reduction (dE* = 0.3). At a higher frequency and temperature, the viscosity decreases, which increases the pigment wetting, Figure 4a. Wetting enables the shear forces produced during extrusion to be transferred onto the pigments to de-agglomerate the particles, which reduces the average pigment size and increases the (frequency) dispersion, Figure 4b, ultimately reducing the color shift (dE*), Figure 4c. Therefore, the shear thinning observed at the center point showed a lower color difference value, indicating an improved pigment dispersion. The correlation between the effects of temperature and rheological characteristics (of the blend) on color shifts has significantly influenced pigment dispersion. For instance, at 255 °C, the pigments were well dispersed for all samples that were processed at 255 °C, 25 kg/h, and 750 rpm (central point). 

#### 3.1.2. Design of Experimental (DoE)—BBD 

The experimental design was considered as follows: extruder FR as the first process parameter, screw Sp. as the second parameter, and heating zone temp as the third parameter. Throughout twelve separate runs, the parameters were varied and extruded so that the impact on color could be investigated in addition to the BBD response method of five center points. To detect the responses’ nonlinearity and provide an estimation of the experimental error, Figure 5 depicts the addition of five center points.

The experimental design took into account the same three process parameters, and the three different levels for each factor coded as −1, 0 and +1, shown in Figure 6. The level used, as well as the extruder FR, Sp., and temp were taken into consideration in the experimental design, as presented in Table 3. 

The L* value for the desired color output was 70.04, the b* value was 18.09, and a* value was 3.41 according to the CIE terms. For statistical data analysis, as part of this study, a confidence interval of 95% was used to compare and examine each factor’s influence on the other components; via the most recent version of the Design Expert Software 8.0. For locating the optimal surface within the given range, a numerical optimization technique was utilized. In response to the design expert, probable ranges of the variables were mapped out. Finally, the data were analyzed using numerical optimization in order to gain color deviations that equaled zero, which is the target color.

### 3.2. Characterization of Polycarbonate Grade Using Spectrophotometry

On a co-rotating intermeshing TSE, extrusion was performed in order to achieve the best homogeneous mixture of the melt. There were three adjustment levels for three distinct parameters. First, the color change effect on screw sp., temp., and FR were explored while keeping the rest fixed. These pellets were then injection molded to three (3 × 2 × 0.1”) color chips (rectangular shape). A total of 85 tons, 1000 PSI and a temp. equal to 280 °C were used in this procedure. A room-temperature drying procedure was used in the laboratory to finish drying the specimens; then they were weighed in three separate places. A spectrophotometer and the CIELAB Color Space system were used to measure and describe the compound color difference tristimulus values (dE*). Colors are measured in terms of the three stimulus values (L*, a*, and b*) and the goal value (Tg) where: CIE L* = 68.5, a* = 1.43, b* = 15.7. Color chips were prepared for spectrophotometer characterization, tin sheets, and microscopy dispersion testing, using a microtome to cut the samples into thin slices. An increase in temperature reduced color differences (dE*), as shown in Table 4. As the temperature range approached 255 °C, the color difference (dE) approached 0.3 and remained so until 280 °C.

### 3.3. Variance (ANOVA) Analysis 

To discover interactions and enhance the processing settings, the researchers utilized an ANOVA to achieve better color quality. The experiment characterized the parameters’ impact on dL*, da*, and db* with the design expert software. To begin, a starting model that was linear in type, then move towards linear, depending on the situation, or quadratic was used. It was clear that quadratic models (0.05 > Prob > F) were the most appropriate models to define dL*, da*, and db* based on the ANOVA findings for the sequential model sum of squares characterizations of these three variables.

A typical R^2^ value for db* would be about 72%, while in our results the value increased to 90% for dL*, da*, as the sum of these three variables is explained. Noise can be a cause of unaccounted for unpredictability in responses. The ”Adjacent R^2^” value, shown in Table 5, seemed to be close to the “R^2^” value, while the signal-to-noise ratio was measured by the ”Adequate Precision” with a preferred ratio > 4, which was the case here.The significant terms for the processing parameters were dL* (A, B, B^2^, BC, C, B^2^), da* (A, A^2^, B, B^2^, C, C^2^, BC, AC) and db*(A, A^2^, B, B^2^, C, BC); where FR = C, speed = B, and temp = A.

### 3.4. Interactions between Process Parameters

#### 3.4.1. Comparison between the Actual and Predicted Values of dE*

Table 6 shows that the differences between the actual and predicted values were relatively small, showing a high degree of agreement. RSM curves facilitated the comprehension of the process conditions as well as determining the parameters’ optimal points. The disparities between the two values were so minimal a significant agreement occurred. 

#### 3.4.2. Interaction of Parameters of da*

Figure 7a shows the interaction contour plot between feed rate and speed (FR–SP) for da* at 274 °C.

The quadratic terms of all three parameters cause the strong curvature that results in elliptical contours. As a result, in two quadrants, the da* value rose as speed and FR increased, whereas in the other two quadrants it fell. Positive interactions existed between FR and speed in two quadrants, whereas negative interactions existed in the other two. 

Figure 7b illustrates the relationship between temperature and FR at 728 rpm. Due to the quadratic connection between all three processing parameters, as shown in Figure 7a, identical elliptic outlines were detected once more. At 728 rpm and 24.4 kg/h yield the best value of da* = 0.20.

Due to the significant influence that FR (C), Sp. (B), and temp. (A) had on dl*, da*, and db*, it was necessary to use a decision-making method that was multi-criteria in nature, as well as applying a total desirability function “d” in order to identify the optimum settings for the three parameters in relation to all responses. 

#### 3.4.3. Parameters Overlay Plot 

An overlay plot of speed response vs. temp contours is shown in Figure 8 when FR equals 24.44 kg per h. This plot illustrates the zone that can be reached to achieve the required values. When da* and db* were equal to 0.197 and −0.188, the contours showed that parameters of temperature and operating speed must meet to satisfy the mean responses (dL*, da*, and db*) when an FR of 24.44 kg per h is kept fixed. For example, at an rpm of 728.38, a speed of 22.44 kg per h, and a temperature of 274.23 °C, the following calculation resulted when dL* equaled −0.01, da * equaled 0.197, and db* equaled −0.188 as tristimulus values. Compared to the highest deviation permitted, which is equal to dE* = 1, the slightest total variation in tristimulus value that can be produced by utilizing Equation (1) was 0.26, which is perfectly acceptable.

In the experimental design, the approach known as the three-level complete factorial response surface BBD was utilized for optimizing the process parameters. In addition, we demonstrated how the processing variables shifted even when the other factors stay the same (general trends (G.T.)). Both approaches resulted in statistically significant changes to the process parameters and a decrease in the color values [28,29].

### 3.5. Characterization and Morphological Dispersion Analysis

The observations in the plastic described pigment dispersion in polycarbonate composites to evaluate and characterize the three processing parameters sample with pigment dispersion. Furthermore, a quantitative methodology was established by combining a 3D X-ray micro-Ct scanner (MCT scanner). X-ray microtomography (XRT) is a technique that uses X-rays to create a three-dimensional image of a sample using a scanning electron microscope [30,31]. 

The previous general trend study discovered that increasing the temperature resulted in the lowest color output variation and decreased agglomerations at 750 rpm, 25 kg/h, and 255 °C (the mid-point between these three parameters). Table 4 demonstrates that dE* = 0.3 within a specified range of process variables was seen simultaneously. When polycarbonate pigment samples were examined using a SEM, a micro-Ct scanner and DOM, they were discovered to be agglomerated, as evidenced by micrograph photographs showing uniform dispersion. De-agglomeration occurred in the high shear zone at the center points due to the increased extrusion parameters of 750 rpm, 25 kg/h, and 255 °C. The material exhibited a reduced shear heat or rate when the process extrusion was set to a low value. As a result, a spherical agglomeration was formed when the shear zones were the lowest.

Researchers performed three separate levels of experimentation on each sample for each of the three parameters. Lower processing parameters resulted in more agglomerated pigment than higher processing parameters in micro-Ct scanner micrographs. Morphological micrographs, aggregation, and pigment dispersion were displayed here in several different characterization modes. The scanning micrographs of injected parts by micro-Ct scanner showed that the distribution improved significantly, similar to the digital microscope and particle size analyzer (previous studies). For samples processed at 255 °C, the pigments were evenly distributed in both cases. Dispersion showed few significant changes, while some agglomerations existed at the same temperature.

Figure 9 shows a consistent distribution of compound polycarbonate pigment samples at the center points (750 rpm, 255 °C, and 25 kg/h). Figure 9a,b show agglomeration and a spherical particle shape, while Figure 9c,d show a uniform pigment distribution. There were no significant changes in dispersion, while a specific number of agglomerates existed at each parameter.

The most common methods for assessing dispersion quality are SEM. Nonetheless, the results obtained here using the techniques indicated that the pigments were in the form of agglomerates and that the color properties of polycarbonate were improved. For example, Figure 10 shows SEM micrographs of the same compound with a relatively large amount of agglomerated pigment at lower processing parameters, more than at higher processing parameters, and significantly less in the center point. In general, the processing conditions at the center point (25 kg/h, 750 rpm, 255 °C) sample demonstrated that the three samples with the slightest color difference were processed more consistently and could be launched as a successfully commercial product for grade compound applications.

The morphology was examined by micro-Ct scanner, digital optical microscopy (DOM), scanning electron microscopy (SEM) and SEM micrographs of the four pigments. The optical microscopic graphs for the processing parameters at higher temperatures, showed the degree of dispersion to be increased. For example, dispersion at 255 °C appeared to be slightly better than at 230 °C. Figure 4 micrographs show that the higher the temperature, the lower the viscosity and evidence of agglomeration. Samples shown in Figure 11 show a DOM micrograph of the compound grade for temperature, feed rate, and speed processing parameters at magnifications ranging from 3000× to 5000×. The pigment particles in this image were from the R1 30% sample. The pigment agglomerated at lower processing temperatures, such as 230 °C. The total color difference, dE*, increased at lower processing parameters, and higher agglomerations than the sample were produced at higher parameters. The SEM micrograph showed the presence of agglomerates in the white, yellow, and red pigments. It revealed spherical and, or elliptical shape in the range of 0.1–0.2 μm. 

Similarly, agglomerates were discovered in the black pigments. Figure 12 depicts SEM images of the black pigments with spherical primary particles with diameters of 10 μm.

Figure 4 shows the temperature effect on color difference (dE*) for the R1 30% sample processed at 255 °C. It shows a consistent decrease in value (dE* = 0.30). Compared to 1.10 at 230 °C, the dE at 280 °C was 0.30 and 0.34 at the central point (255 °C). It shows that the viscosity at 280 °C and 230 °C was sufficient for wetting the particles. The decrease in viscosity and surface tension at higher temperatures of 255 °C and 280 °C can be explained by enhanced pigment wetting compared to 230 °C wetting. The Washburn equation, which is given below, can explain this phenomenon [32].
(2)$$I\left(t\right)=\sqrt{\frac{C\times \bar{r}\times \gamma_{L}\times \cos\theta }{2\times \eta }}\;$$

The equation shows that the rate of pigment wetting depends on the viscosity and surface tension of the fluid. Additives can reduce the dispersion time by reducing the contact angle, as indicated in the Washburn Equation (2), reducing the necessary energy input and preventing re-agglomeration during dispersion. The lower viscosity at 280 °C improved the dispersion process by increasing wetting properties but also enhanced the stability of the pigment concentration in the presence of a stabilizer. In order to finely disperse pigment particles in a liquid, the particles must be ‘wetted’ by the liquid. Air incorporated in the pigment powder must be completely removed, and the pigment particle must be completely surrounded by liquid. The wetting of the pigment particles is influenced by the geometry of the particles, viscosity of the fluid, surface tension, and chemical characteristics of the solvents. However, the breakdown of the pigment particles can occur due to wetting the compound pigments. Producing colored compounds or masterbatches from pigments requires good dispersion by wetting the pigment surface with the polymer, breaking up agglomerates, and separating pigment particles. The better the dispersion of pigment particles, the stronger the color point showed a lower color difference value, indicating improved pigment dispersion. Figure 4 depicts the results of complex viscosity measurements, which show that the compound plastic with higher temp. (280 °C) had lower viscosity than the compound with lower temp (230 °C). The additives and pigments can react with the polymer and soften it, reducing viscosity, accelerating pigment wetting, and improving the color of polymer blend.

### 3.6. The Fourier Transform Infrared Spectroscopy (FT-IR) Analysis

FT-IR technique is beneficial since it allows the identification and localization of compounds to study and identify chemical groups or chemical compounds when the sample absorbs infrared radiation. The amount of residue for the samples produced via the injection molding was significantly lower due to a small ratio of colorants to the total masterbatch weight. It should be noted that, due to the slight usage of samples in TGA (previous study), this method is typically considered a micro-sampling technique and may not reflect the part’s properties as a whole. FT-IR is a method to record the plastic compound’s infrared absorption spectrum. FT-IR can be used to identify altered bands of materials, such as gases, liquids or solids, including polymers. Both pure and mixed materials can have their infrared spectra measured. Therefore, spectroscopic techniques can also be used to study and identify chemical groups or chemical compounds when the sample absorbs infrared radiation. The idea of quantification with a simple measurement (FT-IR or Raman) could help control the composition.

Furthermore, the relationship between viscosity and filler content is that the viscosity always increases with filler addition; the degree of increase depends on the filler geometry [33]. The thermal decomposition of polycarbonate blends occurs through a series of reactions. Therefore, it is giving a weight loss statistic data produced by TGA. the maximum peak temperature is shifted from 430 °C to 580 °C. This significant change in the decomposition temperature results from the delay in the thermal degradation process and is probably due to increased thermal lag. The cause behind this fact is that the sample reaches decomposition temperature in a shorter time when higher heating rates are observed [34].

FT-IR spectra showed the formation of carbonate species CO_2_ at 2347 cm^−1^, the CO_2_ linear coordination produced broad 2300–2400 cm^−1^ IR absorption bands. However, between 1000–1800 cm^−1^, IR-active features were produced by diverse carbonate specie spectra. For example, the C=O stretching carbonate at 1670–1820 cm^−1^, C–O stretching at 1000–1300 cm^−1^, as well as the exhibited 690–900 cm^−1^ C–H ring puckering.

As is shown, the significant development of OH stretching at 3580–3650 cm^−1^ did not appear in the FT-IR curve for the blends with additives and pigments (WA). Instead, FT-IR spectra showed the formation of most functional groups. The peak at 1505 cm^−1^ (para-aromatic ring semi-circle stretching) only existed in the WA blends.

Figure 13 shows the results of the FT-IR analysis of the compound with 30 wt.% of resin R1 (30%/70%) blended with and without additives and pigments (WA and WOA). The main difference between the two procedures was the addition of colorant (WA; WA showed it decreased in absorbance values. Considering these results, it is reasonable to conclude that adding colorant could minimize the viscosity and absorbance mechanisms. The mixture’s viscosity was among the most influential characteristics of pigment dispersion. Low viscosity is required for the fast wetting of pigments. As a result, degradation and yellowing were decreased, and the color difference was reduced, as was seen for the viscosity effects for the polycarbonate blends investigated in the earlier study. Those results suggested that increased peak intensities for OH and CO_2_ in the blends without additives and pigments (WOA) could be attributed to the presence of water or carbon. However, with the pigments and additives (WA), the intensities diminished. With additives, the absorbance was always lowered, the transmittance improved and decreased the plastic sample’s haze. Accordingly, it reduced the color difference (dE*) and improved the quality of the color. The possible existence of water and CO_2_ in the compound plastic without additives was subjected to the performance evaluated as light transmission, haze and color development. For WA, it was found that the additives had a more significant influence on the peak and absorbance performance intensity of the resin grade due to the broader spectral wavelength applied.

## 4. Conclusions

BBD response surface technique and variant processing parameters were used in this investigation to explore the impacts of processing parameters, according to the approach of optimal experimental design. The current study produced models that could predict dL*, da*, and db*. It was found that 274.23 °C, 24.44 kg/h, and 728.38 rpm were the best tristimulus values, with a minimal of 0.26 total deviation, dE*, and general trends at the center point of 255 °C, 750 rpm, 25 kg/h, with a lower color value (dE*) of 0.30. With a very low margin trends in color difference (dE*) in both techniques was about 0.04. As the temperature increased, there was a discernible reduction in the values of color difference (dE*). As a direct consequence of this, it possessed greater peak distribution that was stable throughout. Steadily raising the temperature through a medium range (i.e., with a 255 °C center point) revealed a large minimum color difference. As a result, it revealed a constant higher peak distribution and spherical pigment form. These processing conditions have the potential to create a considerable frictional heat state and shear pressures at higher temperatures. This can have an effect on the pigment’s heat stability and cause damage to the polymer matrix components. De-agglomeration occurred in high shear zones, while dispersion and agglomeration occurred precisely in low shear zones. These discoveries offer an optimal set of processing parameters for polycarbonate grades, reducing color mismatching for both methods. 

As a result, it is reasonable to conclude that different microscopic scanning systems revealed pigment agglomerations in a variety of characterization modes. The pigment clearly agglomerated at lower processing temperatures, less than 230 °C, with spherical or elliptical primary particle shapes. The SEM micrographs showed the presence of agglomerates in the white, yellow, and red pigments with diameters of 0.1–0.2 μm, while SEM images of the black pigments with diameters of 10 μm showed that the black pigments were larger than the other pigments, which could be due to the fact that the black pigments are organic and the other pigments are metallic.

FT-IR revealed the formation of the majority of functional groups. The presence of water or carbon could explain the increasing peak intensities for OH and CO_2_ in blends without additives and pigments (WOA). With additives and pigments, absorbance always decreased. Because of the broader spectral wavelength used, the pigments and additives had a more significant influence on the peak and absorbance performance intensity of the resins grade.

Increasing temperatures and the addition of additives to pigments can soften the polymer, reducing viscosity and surface tension, increasing pigment wetting, decreasing pigment diameters, and thus improving the color of polymer blends. The agglomeration was examined using microscopic scanning and the FT-IR technique was used to study and identify chemical compounds in the sample that absorb infrared radiation. The results of the physical and chemical reactions investigating in these experiments will be correlated with optimal color values (L*, a*, b*) dE*, in order to approach the final research goals. This phenomenon simply indicated that the design color change had a significant effect with reduction the wastage.

## Figures and Tables

**Figure 1 polymers-14-04860-f001:**
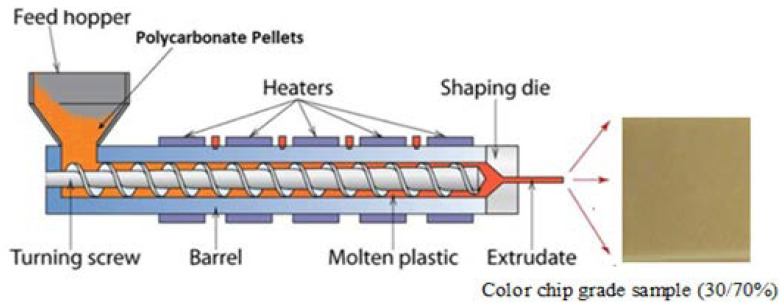
The process of extrusion for a color chip grade sample (30/70%).

**Figure 2 polymers-14-04860-f002:**
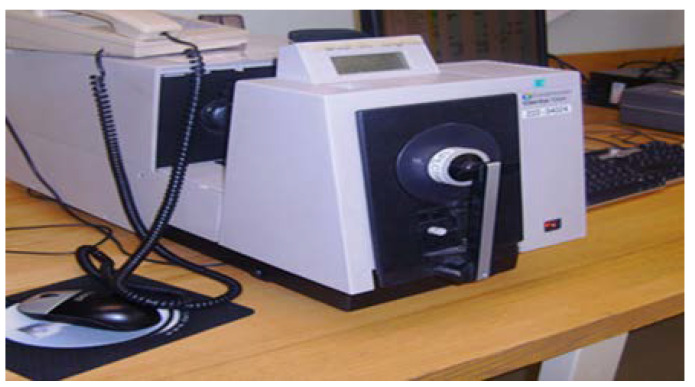
X-rite-7000A spectrophotometer.

**Figure 3 polymers-14-04860-f003:**
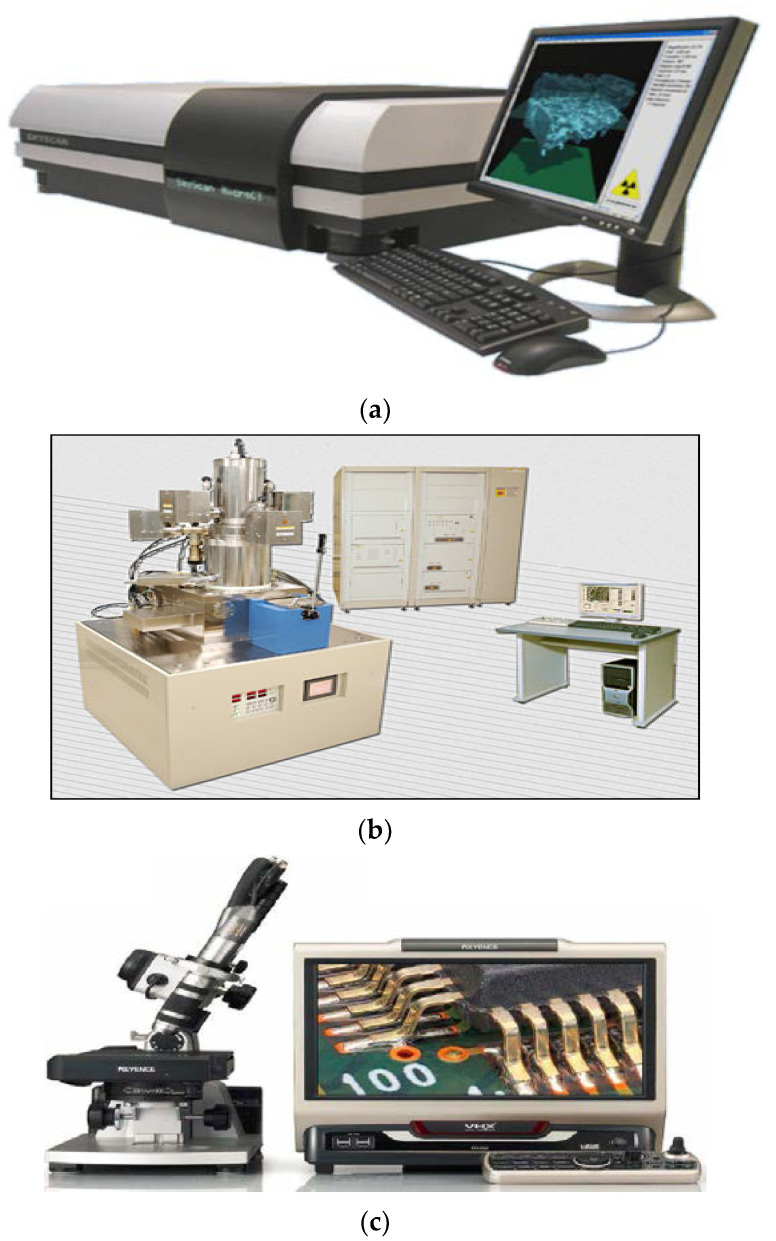
(**a**) A micro-Ct scanner SkyScan 1172, (**b**) Scanning electron microscope (SEM), Joel 5500 LV. (**c**). Digital microscope VHX-1000.

**Figure 4 polymers-14-04860-f004:**
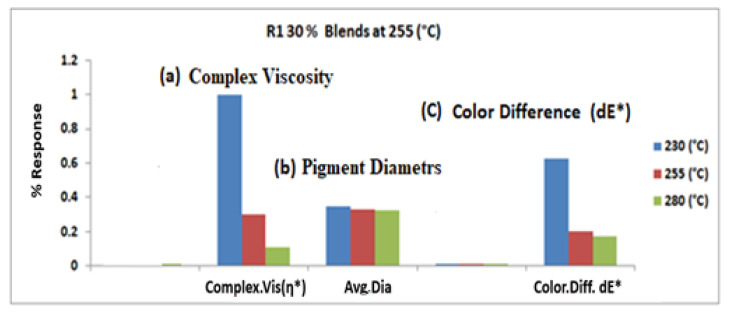
Characterization response at the center point on color.

**Figure 5 polymers-14-04860-f005:**
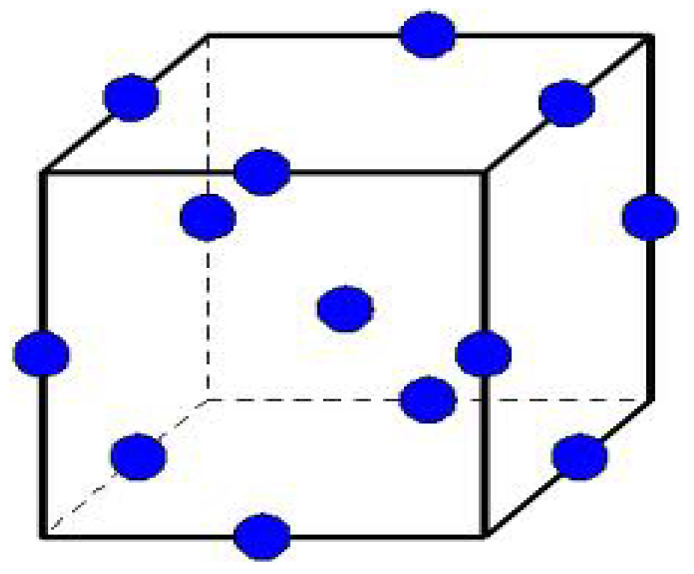
Design of Box–Behnken design (BBD).

**Figure 6 polymers-14-04860-f006:**
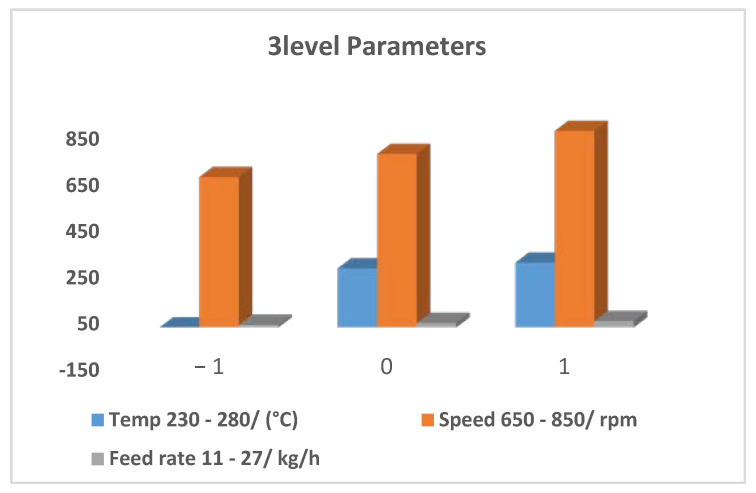
Parameters and experimental design.

**Figure 7 polymers-14-04860-f007:**
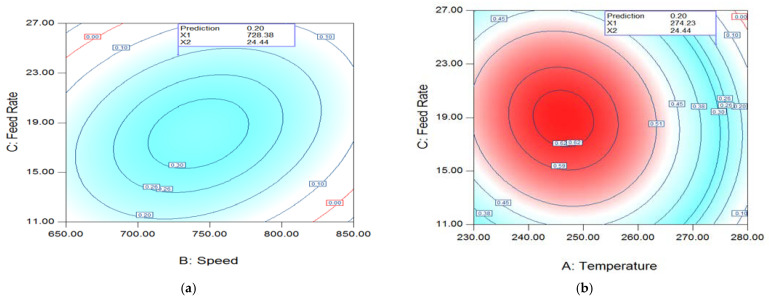
(**a**) da * T, speed, and FR at 728.38 rpm; (**b**) da* temp–FR relationship at 274.23 °C.

**Figure 8 polymers-14-04860-f008:**
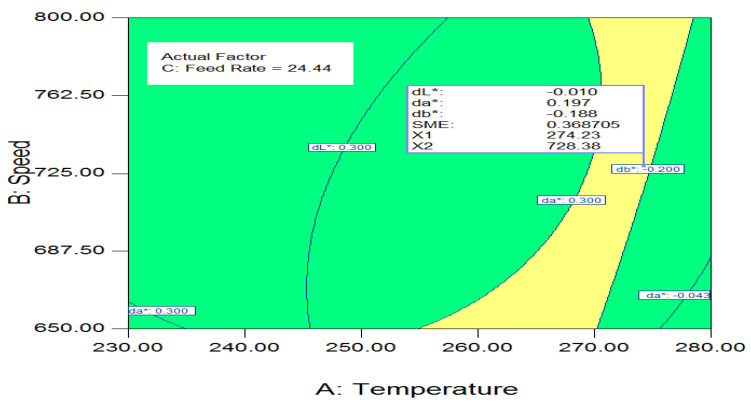
Temperature − Speed overlay plot at 24.44 kg/h FR.

**Figure 9 polymers-14-04860-f009:**
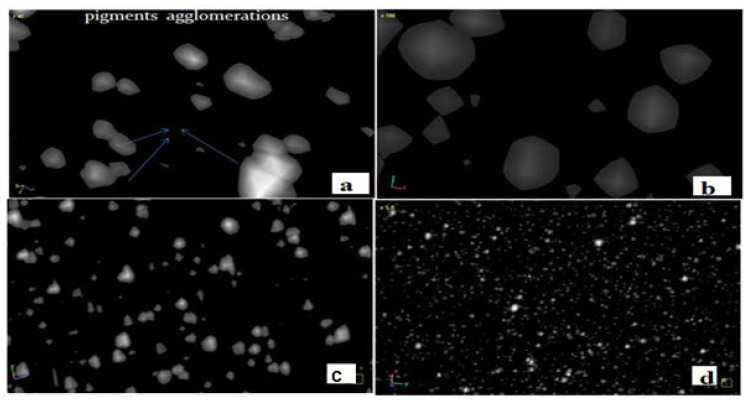
Scanning images of the PC 30/70% blend processed at the center point: (**a**) agglomerations, (**b**) particle shape and (**c**,**d**) pigment distribution.

**Figure 10 polymers-14-04860-f010:**
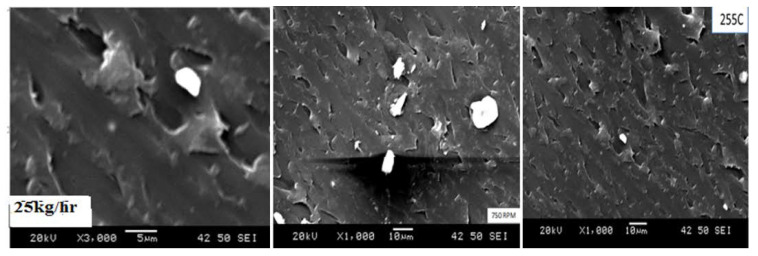
SEM micrographs scanning for the center point for (feed rate, speed, and temp).

**Figure 11 polymers-14-04860-f011:**
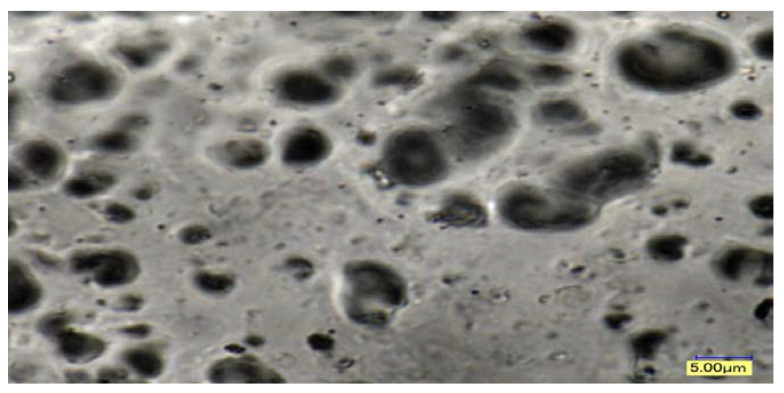
DOM at (5000×) of the PC. Blends at 230 °C.

**Figure 12 polymers-14-04860-f012:**
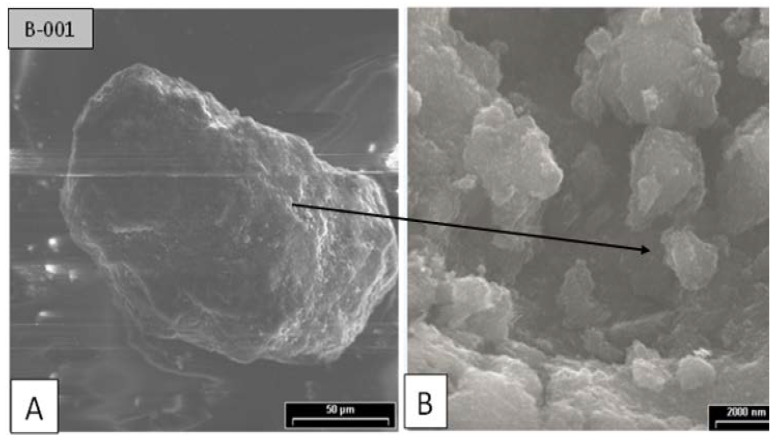
SEM micrograph of black pigments. (**A**) Spherical shape with diameters of 10 μm; (**B**) Agglomerations.

**Figure 13 polymers-14-04860-f013:**
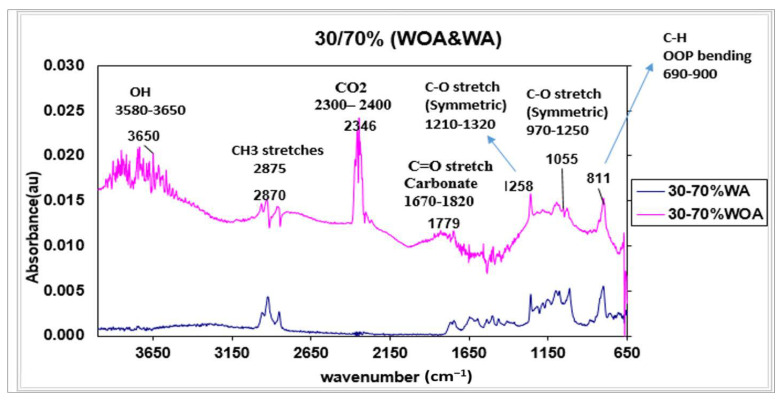
FT—IR spectra of (30%/70%) PC blends. WOA (without additives and pigments) and WA (with additives and pigments).

**Table 1 polymers-14-04860-t001:** Grade color formulation.

#	Ingredients	Name of the Material	Weight	wt/vol
A	Res. (1)	Bisphenol-A	04.9500	grams
B	Res. (2)	Bisphenol-A	10.0500	grams
C	F. (1)	Weather-resist-L	00.0050	milliliters
D	F. (2)	Stabilizer-L	00.0097	milliliters
E	F. (3)	L-Stabilizer	00.0300	grams
F	W	W. pig	00.0416	grams
G	B	B. Pig	00.0050	grams
H	R	R. Pig	00.0262	grams
I	Yell	Yell. Pig	00.0106	grams

**Table 2 polymers-14-04860-t002:** Typical operating processing temperature variations (general trends).

Extrusion Heating Zone Temperature (HZT) (°C)										
Sp.	1	2	3	4	5	6	7	8	9	FR
750	70	195	230	230	230	230	230	230	230	25
750	70	195	255	255	255	255	255	255	255	25
750	70	195	280	280	280	280	280	280	280	25

**Table 3 polymers-14-04860-t003:** Three levels and parameters experimental design.

Processing Parameters	Units of Measurement	Three Code Levels		
		−1.0	0.0	+1.0
T.	degrees Celsius	230	255	280
S.	revolutions per minute	650	750	850
F. R.	kilogram per hour	11	19	27

**Table 4 polymers-14-04860-t004:** G.T. extruded processing parameter effect at the center points.

# of Run	Speed.RMP.	T (°C)	FR.kg/h.	L.*	a.*	b.*	dE.*
(1)	750	230	25	067.91	1.400	14.760	1.100
(2)	750	240	25	068.62	1.520	15.403	0.320
(3)	750	255	25	068.42	1.470	15.350	0.350
(4)	750	270	25	068.43	1.320	15.460	0.320
(5)	750	280	25	068.66	1.520	15.440	0.300

The L*-axis represents lightness and ranges from 0 (black) to 100 (White). The other two coordinates, a* and b*, respectively, represent redness-greenness and yellowness-blueness.

**Table 5 polymers-14-04860-t005:** Analysis of variance for tristimulus color values.

Tristimulus Values	Adequate Precision	Predicted R^2^	Adjacent R^2^	R^2^
dL*	17.42	0.84	0.91	0.94
da*	27.8	0.89	0.97	0.98
db*	5.62	0.40	0.56	0.72

The dL*, da*, and db* values provide a comprehensive numerical description of the color differences between a Sample and a Standard color. dL* denotes the difference in lightness between the sample and standard colors. da* denotes the difference in redness or greenness between the sample and standard colors. db* represents the difference in blueness and yellowness between the sample and standard colors. The customer usually sets the permissible tolerance limits in terms of dL*, da*, db*, or dE*; however, for the polycarbonate grade under study, limits were ≤ 0.6 for dL*, da*, db* and ≤ 1.0 for dE*.

**Table 6 polymers-14-04860-t006:** Actual and predicted values comparison of dE*.

Optimizing-Runs	Minimum Color Values (dE*)	
	Actual Value	Pred.-Value
1	0.380	0.249
2	0.381	0.476
3	0.789	0.652
4	0.345	0.200
5	0.291	0.393
6	0.815	0.652
7	0.592	0.652
8	0.651	0.705
9	0.344	0.299
10	0.601	0.652
11	0.700	0.623
12	0.592	0.690
13	0.708	0.652
14	0.648	0.685
15	0.480	0.500
16	0.353	0.278
17	0.664	0.625

dE*: A symbol used to indicate deviation or difference. The total color difference computed with a color difference equation. dE* is a single number or a geometric distance between two points in three-dimensional space that represents the color difference between two readings and is based on the L*, a*, and b* color space system.

## Data Availability

The data presented in this study are available on request from the corresponding author.

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
