# Peer review of "Box–Behnken Design for Polycarbonate-Pigment Blending: Applications and Characterization Techniques"

_polymers, 2022, doi:10.3390/polym14224860_

Round 1

Reviewer 1 Report

In this manuscript, the authors report during polymer compounding extrusion, three effective processing factors were investigated: feed rate (FR), Speed (Sp.) and temperature (Temp.) for a coloring compounded polycarbonate grade (30/70%). This processing design techniques were obtained by employing the application of design experiment in response surface methodology (RSM) for blending two polycarbonates with pigments and optimizing processing temperatures at center points. The article shows some valuable results. However, there are still some mistakes and shortcomings in the article, which need to be major revised carefully before it can be accepted by Polymers. Specific comments are described below:

1. Please check the entire manuscript to improve the clarity and accuracy of your writing. Here are some examples: Page 2, line 89 the four pigments written in the manuscript, why there are five pigments in table 1, please explain.

2. The language of the paper needs thorough revision. There are many grammatical and syntax errors, which must be corrected. On page 3, line 112, ‘in’ to be changed to ‘In’; On page 3, line 115, ‘R 1’ to be changed to ‘R1’.

3. On page 4, line 139, ‘was’ to be changed to ‘were’. On page 4, and line 139 to146, the sentence ‘While…..in Table.2 .’ Is to be re-written.

4. There are so many tables in the manuscript, it is suggested that some secondary tables could be put into supporting information.

5. Reading through the whole manuscript, the author did not use any chemical methods such as FT-IR and XPS to prove whether the polymer and the pigments reacted. It is recommended that the author added.

6. In references, the reference format needs to be arranged according to the journal requirements.

7. Fig. 8 and 9 are not clear enough.

Author Response

Hello,

I am writing to you regarding the [Manuscript ID: polymers-1943919] entitled " Box-Behnken Design for Polycarbonate-Pigment Blending: Applications and Characterization Techniques " which was submitted to MDPI-Polymer.

I submitted my article on September 15, 2022; I presented my study on a novel technique for enhancing designing material with pigment characterization using different techniques - characterization.

I have received the reviewer's comments, which are highly important and valuable. I have handled all of the following comments thoroughly and fixed them as necessary, and I appreciate the reviewers' guidance:

  • I have fixed all grammar mistakes throughout the article. I have reviewed my article once again and consistently go through in describing the images. I have considered professionally edited for the English language. Formatting the template …etc.
  • I have presented more literature, topics, figures, tables, equations, and what other researcher in this area has done.
  • I made my research paper clear and easy to understand.
  • Finally, I have included additional journal references, generated substantial findings, and revised and responded to all questions and comments in this paper.
  • The article provides detailed answers to the reviewer's questions.

The First reviewer comments:

  1. Please check the entire manuscript to improve the clarity and accuracy of your writing. Here are some examples: Page 2, line 89 the four pigments written in the manuscript, why there are five pigments in table 1, please explain.

There are four pigments, three additives, and two resins; however, to make it more clearly, I switched to the material formulation in terms of weight/volume rather than PPH, as shown in Table 1 in this paper.

  1. The language of the paper needs thorough revision. There are many grammatical and syntax errors, which must be corrected. On page 3, line 112, ‘in’ to be changed to ‘In’; On page 3, line 115, ‘R 1’ to be changed to ‘R1’.

I went through the paper and corrected where necessary, changing in to in and R 1 to R1.

  1. On page 4, line 139, ‘was’ to be changed to ‘were’. On page 4, and line 139 to146, the sentence ‘While…..in Table.2 .’ Is to be re-written.

I changed was to were; and from lines 139 to 146, the sentence was rewritten in a very clear way, from while...to till

  1. There are so many tables in the manuscript, it is suggested that some secondary tables could be put into supporting information.

To supplement our information, additional figures/models/Tables, topics (such as Rheology, SEM, DOM, FT-IR and Colors), equations and references.

  1. Reading through the whole manuscript, the author did not use any chemical methods such as FT-IR and XPS to prove whether the polymer and the pigments reacted. It is recommended that the author added.

Throughout the manuscript, I used various chemical methods Rheology, spectrophotometer, CT scanner SEMs and DOM. Also, I used the FT-IR (without pigment and additives (WOA) and with pigment and additives (WA) and scanned the results with a micro-CT scanner and SEM to confirm the dispersion effect and reaction on color. Results of FT-IR approve a significant reaction of the polymer with the pigment and additive.

  1. In references, the reference format needs to be arranged according to the journal requirements.

I arranged the references format according to the journal templates.

  1.  Fig . 8 and 9 are not clear enough.

I redrew a clear CT scanner figure and included it with a new SEM, DOM micrographs, and FT-IR.

I appreciate your time and constructive comments.

Best Wishes

Reviewer 2 Report

Comments: 

1)Title : very long

2) Abstract:  fewer agglomerates...... have a good insight (could you please add percentages)

3) Keywords : OK

4) Introduction: reference [3] is missing , please add more references to show the art state of your subject

5) Materials and Experiments: Table 1 (0.20; 0.05......)

Figure 1, please put a and b on the design

6) Results and Discussion: what happen to your materials  below 230°C

Table 4 plitted betwen two pages

Figure 8 : explain what is a and b

Figure 9: put a and b too

7) Conclusion : ok

8) References : add more and update them by adding 2022.....

with regards

Author Response

Hello,

I am writing to you regarding the [Manuscript ID: polymers-1943919] entitled " Box-Behnken Design for Polycarbonate-Pigment Blending: Applications and Characterization Techniques " which was submitted to MDPI-Polymer.

I submitted my article on September 15, 2022; I presented my study on a novel technique for enhancing designing material with pigment characterization using different techniques - characterization.

I have received the reviewer's comments, which are highly important and valuable. I have handled all of the following comments thoroughly and fixed them as necessary, and I appreciate the reviewers' guidance:

  • I have fixed all grammar mistakes throughout the article. I have reviewed my article once again and consistently go through in describing the images. I have considered professionally edited for the English language. Formatting the template …etc.
  • I have presented more literature, topics, figures, tables, equations, and what other researcher in this area has done.
  • I made my research paper clear and easy to understand.
  • Finally, I have included additional journal references, generated substantial findings, and revised and responded to all questions and comments in this paper.
  • The article provides detailed answers to the reviewer questions.

1) Title: very long

The new Title:

Box-Behnken Design for Polycarbonate-Pigment Blending: Applications and Characterization Techniques.

2) Abstract:  fewer agglomerates...... have a good insight.

The output response might be significantly impacted when excellent coloring dispersion was observed with fewer agglomerates; the average pigment size measured approximately 0.1-0.2 μm for white, yellow, and red pigment and 10 μm for black pigments.

The optimal number of particles was increased when additives were used with pigments, and the average particle size decreased to approximately0.1- 0.2 µm. It was also observed that agglomeration occurred in zones of high pigment size and low processing temperatures.

The SEM micrograph depicts the presence of agglomerates in white pigments. It reveals the existence of primary particles that had a spherical shape in the vicinity of 0.1 mm in size. Also, it shows SEM image of yellow pigments. The figure shows agglomerates consisting of primary particles of elliptical or cylindrical shapes with a diameter of approximately 0.1-0.2 mm. Similarly, agglomerates were found in black and red pigments having primary particles in a spherical shape with diameters of 10 mm and 0.1 mm.

Inadditions.I have used more scanning system to characterized the agglomeration in different ways (CT scanner, SEM, DOM, FT-IR and rheology)

3) Keywords: OK

4) Introduction: reference [3] is missing, please add more references to show the art state of your subject

I have added additional references and more figures, tables, and references to show the state of the subjects:

Processing, dispersion, characterizations, and microscopic scanning (SEM, DOM Rheology, and FT-IR) are all part of materials.

Revise the abstract, the introduction, the results, and the conclusions. 5) Materials and Experiments: Table 1 (0.20; 0.05......).

There are four pigments, three additives, and two resins; the three additives are in terms of volume and weight; however, to make it more clear, I switched to the material formulation in terms of weight/volume rather than PPH, as shown below.

Table 1: Grade's Color Formulation (enclosed /attached article)

Figure 1, please put a and b on the design.

I have modified the diagram and removed a and b with one design

6) Results and Discussion: what happen to your materials below 230°C.

The pigment clearly agglomerated at lower processing temperatures, such as 230°C. The total color difference, dE*, increased at lower processing parameters. Higher agglomerations than the sample were produced at higher parameters

which decreases pigment wetting, decreases wetting disables, and reduces the shear forces produced during extrusion to be transferred onto the pigments in order to de-agglomerate particles, which increases average pigment size and decreases the (frequency) dispersed, ultimately increasing the color shift (dE*). Therefore, the shear thinning observed at the center point showed a higher color difference value, indicating affect the pigment dispersion negatively. The correlation between the effects of temperature and rheological characteristics (of the PC blends) on color shifts has been shown to significantly influence pigment dispersion.

Table 4 splitted between two pages.

I have plotted the table in one page.

 Figure 8: Explain what is a and b?

Replace the figure with clearer one and explained each part (a, b, c, and d)

Figure 9: Put a and b too.

To make it more clear. I plotted with four images and high resolutions. Then, I plotted just one figure and marked it with the letters a, b, c, and d. Also, I have plotted more scanning micrographs and created additional figures such as 9, 10, 11, 12, and 13.

7) Conclusion: Ok.

8) References: Add more and update them by adding 2022.

I added more comprehensive, recent articles, and references related to my research.

I appreciate your time and constructive comments.

Best Wishes

Round 2

Reviewer 1 Report

The author of the article has revised it as requested and recommends acceptance.